# Advancing stream classification and hydrologic modeling of ungaged basins for environmental flow management in coastal southern California

Stephen K. Adams[1], Brian P. Bledsoe[2], Eric D. Stein[3]

[1]Department of Civil and Environmental Engineering, Colorado State University, Fort Collins, CO, 80523, USA
[2]Institute for Resilient Infrastructure Systems, College of Engineering, University of Georgia, Athens, GA, 30602, USA
[3]Southern California Coastal Water Research Project, Costa Mesa, CA, 92626, USA

*Correspondence to*: Stephen K. Adams (Stephen.Adams@colostate.edu)

**Abstract.** Environmental streamflow management can improve the ecological health of streams by returning modified flows to more natural conditions. The Ecological Limits of Hydrologic Alteration (ELOHA) framework for developing regional environmental flow criteria has been implemented to reverse hydromodification across the heterogenous region of coastal southern California (So. CA) by focusing on two elements of the flow regime: streamflow permanence and flashiness. Within ELOHA, classification groups streams by hydrologic and geomorphic similarity to stratify flow-ecology relationships. Analogous grouping techniques are used by hydrologic modelers to facilitate streamflow prediction in ungaged basins (PUB) through regionalization. Most watersheds, including those needed for stream classification and environmental flow development, are ungaged. Furthermore, So. CA is a highly heterogeneous region spanning gradients of urbanization and flow permanence, which presents a challenge for regionalizing ungaged basins. In this study, we develop a novel classification technique for PUB modeling that uses an inductive approach to group perennial, intermittent, and ephemeral regional streams by modeled hydrologic similarity followed by deductively determining class membership with hydrologic model errors and watershed metrics. As a new type of classification, this "Hydrologic Model-based Classification" (HMC) prioritizes modeling accuracy, which in turn provides a means to improve model predictions in ungaged basins, while complementing traditional classifications and improving environmental flow management. HMC is developed by calibrating a regional catalog of process-based rainfall-runoff models, quantifying the hydrologic reciprocity of calibrated parameters that would be unknown in ungaged basins, and grouping sites according to hydrologic and physical similarity. HMC was applied to 25 USGS streamflow gages in the south coast region of California and was compared to other hybrid PUB approaches combining inductive and deductive classification. Using an Average Cluster Error metric, results show HMC provided the most hydrologically similar groups according to calibrated parameter reciprocity. Hydrologic Model-based Classification is relatively complex and time-consuming to implement, but it shows potential for simplifying ungaged basin management. This study demonstrates the benefits of thorough stream classification using multiple approaches, and suggests that Hydrologic Model-based Classification has advantages for PUB and building the hydrologic foundation for environmental flow management.

**1 Introduction**

The natural variability of streamflow regimes, including flow magnitude, duration, frequency, timing, and rate of change (Poff et al., 1997), is crucial for maintaining the ecological integrity of streams (Bunn and Arthington, 2002). Maintenance of aquatic and riparian ecosystem functions is a major priority for water managers; however, streamflow regimes have been altered globally as population growth and development lead to urbanization, dams, flow extraction, and other land use changes (Naiman et al., 1995; Richter et al., 1997).

Issues of ecological integrity are particularly pronounced in intermittent and ephemeral streams located in arid and semi-arid regions, where the spatial variation of flow regimes and ecological functions are less understood than more humid environments (Skoulikidis et al., 2017; Stubbington et al., 2018). Historically, non-perennial streams have been thought of as biologically inactive with poor biodiversity, resulting in devalued ecosystems. These incorrect assumptions have deemphasized research of intermittent and ephemeral streams in favor of perennial streams (Datry et al., 2014). As the importance of healthy non-perennial stream ecosystems has come into focus, a better understanding of requisite hydrologic processes has been established. Generally, non-perennial streams are characterized by sparse and variable precipitation with minimal groundwater influence (Tooth, 2000). These climactic conditions result in streams that are often dry, but highly flashy (Gannon et al., 2022). As such, most studies on physical processes have occurred when these typically dry channels are wet (Tooth, 2000). A large study spanning 540 watersheds of non-perennial streams across the US identified three flow metrics that regionally separate non-perennial streams: No-flow fraction, First day with no flow, and Days between peak and no flow (Hammond et. al, 2020).

Environmental flow criteria frameworks, such as the Ecological Limits of Flow Alteration (ELOHA) (Poff et al., 2010), are methods for protecting the ecological health of streams from hydrologic alteration by reestablishing essential elements of streamflow and sediment regimes. The ELOHA framework is robust because it synthesizes many flow-ecology relationships from a study area to provide a foundation for developing environmental flow recommendations within an entire municipality or management region (Poff et al., 2010). Such a regional approach has been recommended for the widespread implementation of environmental flows because it allows for effective and comprehensive estimation of environmental streamflow regimes at a wide variety of streams in a large and diverse study area (Arthington et al., 2006). The coastal area of southern California (So. CA) is a semi-arid region experiencing substantial hydrologic alteration (Hawley and Bledsoe, 2011) and associated ecological decline (Stein et al., 2012), which has prompted application of ELOHA (Mazor et al., 2018; Parker et al., 2019; Pyne et al., 2017; Sengupta et al., 2018; Stein et al., 2017). The region is highly heterogenous, spanning an extensive range of geology, stream types, and land uses, which presents unique challenges for implementing ELOHA.

Stream classification is one of four major steps within the scientific process of ELOHA used to group hydrologically, or otherwise similar, streams (Poff et al., 2010). Its primary role towards developing environmental flows is to stratify flow-ecology relationships by regional stream type, and to help determine where new bioassessment sites should be placed to strengthen the variety of sites within a region. Olden et al. (2012) outlined two overarching approaches to hydrologic classification—those utilizing inductive reasoning (observed or modeled flows) and those utilizing deductive reasoning

(watershed data characterizing flow). While the inductive approach benefits from actual measures of discharge, it is often plagued by insufficient gauging networks (Olden et al., 2012) and uncertainty modeling ungaged basins (Blöschl et al., 2013). These challenges are particularly prevalent in arid and semi-arid regions where gage records are limited (Merritt et al., 2021) and modeling methods were not developed (Costigan et al., 2017). Despite these obstacles, Merritt et al. (2021) used inductive hydrologic classification to group 287 stream reaches in the arid and semi-arid western US. Metrics describing zero-flow conditions were the strongest class predictors.

Two mirroring state-wide stream classification studies utilizing both inductive and deductive approaches have recently been performed across California (CA). Pyne et al. (2017) first clustered all stream reaches based on similarity of watershed characteristics, then used hydrologic metrics to determine cluster membership and separate reference reaches. Conversely, Lane et al. (2017) grouped unimpaired gages based on their natural streamflow regime before using watershed characteristics to predict the flow type of ungaged reaches. A third state-wide classification study was performed by Lane et al. (2018), which unified the classifications of Pyne et al. (2017) and Lane et al. (2017) by using daily-scale hydrologic baseline archetypes based on dimensionless reference hydrographs. These three stream classification studies focused on characterizing natural flow regimes across California, which is a challenge in the heavily hydrologically modified and heterogeneous Southern Coast hydrologic region of CA (Waananen and Crippen, 1977). Sites from this region did not show strong separation from the rest of CA in previous classifications. While most South Coast streams were classified as "rain and seasonal groundwater" (Lane et al., 2017) or "rain and seasonal groundwater" and "flashy, ephemeral rain" (Lane et al., 2018), not one of the 91 reference gages used to drive the Lane et al. (2017) classification fell in the South Coast. Furthermore, streams in the Mohave Desert and Central Valley shared the same "rain and seasonal groundwater" classification and South Coast streams (Lane et al., 2017). Central Valley streams remained grouped with South Coast streams in the unified classification (Lane et al., 2018). Finally, none of the seven classes produced by Pyne et al. (2017) were dominated by South Coast streams. The results of these three state-wide classifications indicate developing environmental streamflow criteria for South Coast streams could benefit from a more targeted classification focused on the diverse regional landscape.

Regionalization is a common framework for predicting streamflow in ungaged basins (PUB) that is performed by transferring hydrologic information from gaged systems to ungaged (Blöschl et al., 2013; Razavi and Coulibaly, 2013). While regionalization often employs regression equations to compute a single streamflow metric, such as peak flow, conceptual hydrologic models offer continuous process-based analyses with full hydrograph outputs that can be used to analyze past and future climate, land use, and management scenarios. The application of hydrologic models to these alternative scenarios makes them important for developing the hydrologic foundation within ELOHA (Poff et al., 2010). Additionally, a hydrologic foundation often necessitates modeling of ungaged basins because crucial bioassessment sites used to develop flow-ecology relationships often occur on small streams without available representative streamflow data (Poff and Ward, 1989). While modeling ungaged basins for general hydrologic analyses may focus on different flow characteristics and process than modeling for developing environmental flow standards, the importance of PUB to ELOHA and other stream management efforts is clear, and yet no superior method for regionalizing hydrologic models has emerged (Blöschl et al., 2013).

In a typical flow regionalization effort with hydrologic models, many models are created and calibrated at gaged sites across a study area. For ungaged sites within the study area, model parameters that cannot be calculated directly are estimated and/or transferred from the catalog of calibrated models, typically using a measure of spatial proximity, physical/hydrologic similarity, or parameter regression (Oudin et al., 2008; Razavi and Coulibaly, 2013; Samuel et al., 2011). While spatial proximity is generally the preferred regionalization approach (Razavi and Coulibaly, 2013), it is not always superior and is less applicable in highly heterogeneous regions, such as So. CA, where neighboring watersheds may have substantially different geology, land use, and/or climate. Regionalization can also utilize regression equations to directly estimate calibrated parameters (Abdulla and Lettenmaier, 1997; Seibert, 1999; Yokoo et al., 2001); however, this approach handles each parameter individually and does not account for interactions between them (Oudin et al., 2008). These challenges with applying a traditional regionalization approach in a highly heterogenous region provide opportunities for PUB innovations, such as recent developments with random forest models (Prieto et al., 2019) and regionally trained long short-term memory (LSTM)-type models (Kratzert et al., 2018), which are a type of neural network. Furthermore, the technique of grouping similar streams is shared by ELOHA and PUB, which provides an excellent opportunity to explore new approaches for classifying streams with the intention of modeling ungaged basins while developing environmental flow criteria in a highly heterogeneous region.

This study was motivated by a desire to improve the science supporting environmental streamflows in So. CA where flow criteria are under development (Mazor et al., 2018; Parker et al., 2019; Sengupta et al., 2018; Stein et al., 2017) and management of ephemeral streams is challenging (Chiu et al., 2017). In this study, we develop a new method of stream classification that quantifies hydrologic similarity for regionalizing ungaged basins in a heterogeneous region. We compare this new approach to traditional methods of stream classification using hydrologic and watershed characteristics. Towards this end, this study has three specific objectives:

1)      Classify streams in coastal southern California using the existing approaches;

2)      Develop and implement a new approach for stream classification that prioritizes the accuracy of regional hydrologic models; and

3)      Compare the accuracy of traditional classifications versus the new approach for estimating streamflow and flow-ecology relationships in heterogeneous ungaged basins.

We hypothesize that directly incorporating regional model accuracy into a stream classification scheme will provide information complementary to existing deductive and inductive schemes and demonstrate greater ability to accurately model ungaged basins through regionalization, compared to the traditional classifications.

## 2 Methods

### 2.1 Study Area

This study was focused within the large coastal region of southern California, which is roughly bounded by the transverse mountain ranges to the north, Mexico to the south, the peninsular mountain ranges to the east, and the Pacific Ocean

to the west. Study watersheds lie within the coastal regions of San Diego, Riverside, Orange, San Bernardino, Los Angeles, Ventura, and Santa Barbara Counties, and are considered within the "South Coast" hydrologic region of CA according to the U.S. Geological Survey (USGS) (Waananen and Crippen, 1977). The climate is characterized as semi-arid and Mediterranean with hot, dry summers and mild, wet winters. Diverse regional topography, geology, and precipitation patterns allow for the natural existence of many stream types, spanning perennial, intermittent, and ephemeral. Land use varies dramatically across

the region ranging from heavily urban and suburban sprawl, to significantly agricultural, to rural coastal and mountainous. These diverse land uses profoundly influence streamflows, with particular deviation from natural flow regimes occurring due to the urban centers of Los Angeles and San Diego concurrently with the California State Water Project.

        As a first step towards developing environmental flow criteria, only USGS stream gage sites were considered with neighboring bioassessment sites from the California Water Boards' Perennial Streams Assessment (PSA) within the Surface

Water Ambient Monitoring Program (SWAMP). This provided gaged flow estimates at bioassessment sites. Hydrologic surrogacy between gage and bioassessment sites was assumed by ensuring a difference in watershed area of less than 15% with no intervening dams, diversions, reservoirs, or interbasin transfers. Gages from the region were selected to contain high-resolution hourly streamflow data for water years (WY) 2005-2007, which typify relatively wet, average, and dry years consecutively in So. CA (WRCC, 2015). Finally, watersheds of selected gages required sufficient meteorological and

landscape data to build minimally calibrated rainfall-runoff models (Sect. 2.3.1). An exhaustive search for suitable streamflow records yielded 25 USGS gage sites for classification (Fig. 1; Table A1).

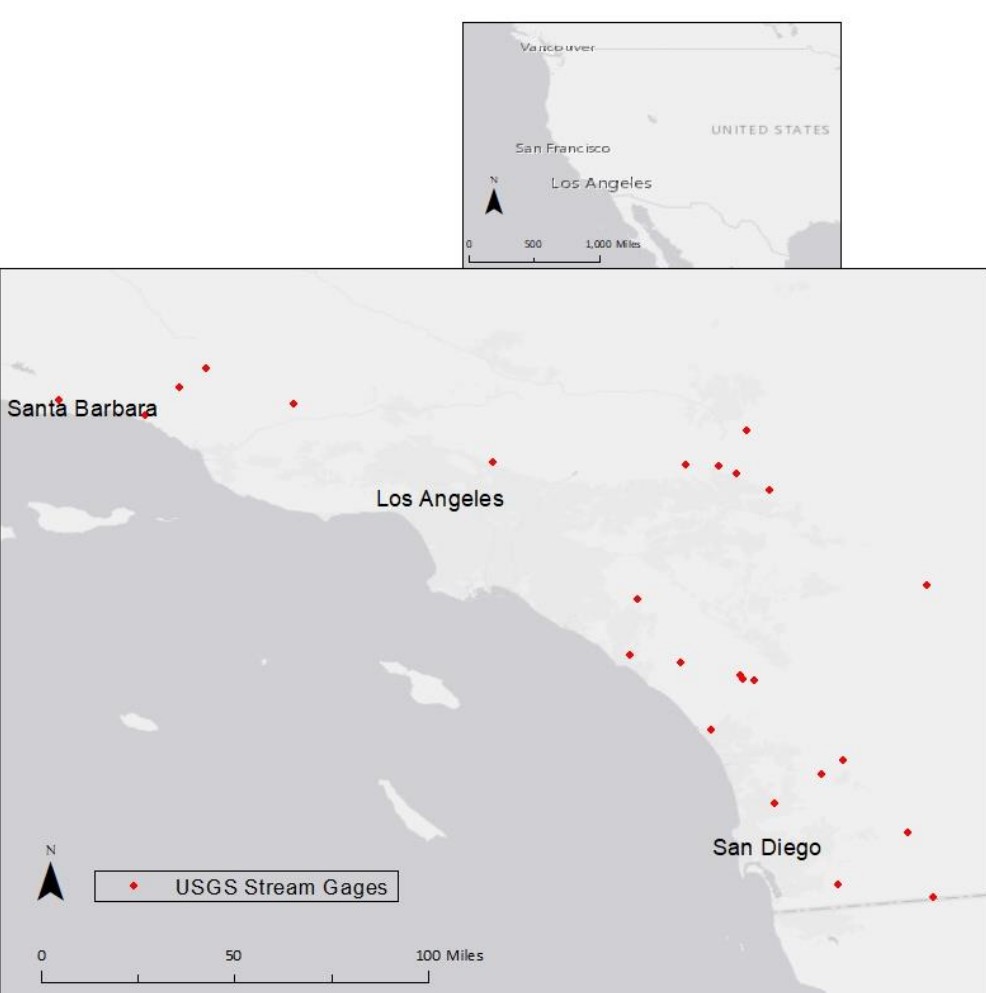

**Figure 1: Locations of USGS streamflow gages used for classification.**

## 2.2 Traditional Classification

Three types of traditional classification were used in this study: an inductive approach with gaged flow data, a deductive approach utilizing watershed characteristics, and a combined inductive and deductive approach applying both types of data.

### 2.2.1 Inductive Approach

        Research in So. CA has shown streamflow flashiness and drying have important influence on shaping local benthic
macroinvertebrate assemblages and ecosystem health (Gasith and Resh, 1999). While flood and average flow conditions play roles in shaping ecological health in arid regions (Merritt et al, 2017; Yarnell et al., 2020), preceding ELOHA studies in So.

CA have found stronger relationships with flashiness and stream drying (Mazor et al., 2018; Parker et al., 2019). To this end, flashiness and drying have been extensively studied as management endpoints for developing regional environmental flow criteria (Mazor et al., 2018; Parker et al., 2019; Pyne et al., 2017; Sengupta et al., 2018; Stein et al., 2017). This study builds upon this foundation by analyzing Richards-Baker Flashiness Index (RBI) (Baker et al., 2004) and a metric quantifying the frequency of extremely low flows indicative of drying were computed from the 25 hourly time series of discharge. RBI was calculated according to Eq. 1, wherein $Q_t$ is the discharge at time t, $Q_{t+1}$ is the discharge at time step after t, and T is the final time step.

$$RBI = \frac{\sum_{t=1}^{T}|Q_{t+1}-Q_t|}{\sum_{t=1}^{T}Q_t} \tag{1}$$

To quantify the frequency of extremely low flows indicative of drying, the fraction of flow record with flow less than 1 cfs was calculated according to Eq. 2, wherein $N_{Q<1cfs}$ is the number of time steps containing streamflow less than 1 cfs and N is the total number of time steps containing flow data. This metric is essentially the same as the No-flow fraction from Hammond et. al, (2020).

$$< 1\,cfs = \frac{N_{Q<1cfs}}{N} \tag{2}$$

Although flows less than 1 cfs are recorded by USGS, this threshold was chosen instead of 0 cfs to indicate stream drying given the inherent measurement error associated with stream gage data at extreme low flows. Due to So. CA's heterogeneous landscape, large variations in land use, topography, and precipitation shape flow permanence and flashiness across the region (Table A1). To better discern the effects of these heterogeneities on streamflow, and to more accurately capture time-sensitive environmental flow metrics on a scale relevant to benthic macroinvertebrates, hourly data were chosen over daily. Additionally, high resolution hourly data across So. CA provide an opportunity to complement the previous state-wide classifications (Lane et al., 2017; Lane et al., 2018; Pyne et al., 2017), which used daily average streamflow data, at finer temporal and spatial scales.

Inductive classification was performed to group sites based on similarity of streamflow flashiness (RBI) and permanence (< 1 cfs). To achieve this, a variety of exploratory ordination analyses were conducted to develop an initial understanding of how gages might classify. Principal component analysis (PCA) was first used to assess linear relationships between flow metrics at the 25 sites (R Core Team, 2019), while weighted classical (metric) multidimensional scaling analyzed non-linear relationships (Oksanen et al., 2019). Classification was ultimately determined using K-means clustering (Charrad et al., 2014) with Euclidean distance. Indices that measure distances between and among clusters (C-Index, Dunn, McClain, and Silhouette) were considered in conjunction with exploratory analyses to determine the number of clusters. K-means clustering with Euclidean distance is a robust approach that does not depend on the statistical distribution of data (Hartigan and Wong, 1979). It is one of the most common and well-established self-learning clustering algorithms.

**2.2.2 Deductive Approach**

For traditional deductive classification, watershed data describing USGS streamflow gages were retrieved from the USGS's GAGES-II database (Falcone, 2011) and the U.S. Environmental Protection Agency's (EPA) NHDPlusV2 database (McKay et al., 2012). Correlation was performed (R Core Team, 2019) to reduce the large pool of watershed metrics. If two metrics contained a correlation coefficient > 0.5, then they were considered highly correlated, and one was removed. Judgement was applied to include more general metrics in favor of metrics with greater specificity (i.e., remove "Mean Jan Precip" and do not remove "Mean Annual Precip"). Finally, the same exploratory ordination analyses and clustering process as the inductive approach provided results for traditional deductive classification.

**2.2.3 Combined Inductive and Deductive Approaches**

Inductive and deductive methods of stream classification were combined in multiple ways. First, a single K-means clustering analysis was performed using the hydrologic metrics (RBI and < 1 cfs) and the best performing watershed variables from the deductive classification. Next, multinomial logistic regression (Venables and Ripley, 2002) was used to determine if flow metrics could predict deductively produced clusters, and likewise used to see if landscape metrics could predict inductively produced clusters. Finally, the USGS has categorized streamflow gages containing minimally disturbed watersheds without significant flow alteration as "reference" within the GAGES-II database (Falcone, 2011). Multinomial logistic regression with flow and watershed metrics was again used to predict whether a gage was reference or non-reference.

**2.3 Hydrologic Model-based Classification**

Hydrologic Model-based Classification (HMC) first requires the accurate creation and calibration of rainfall-runoff models across a region, exactly like regionalization for estimating streamflow in ungaged basins. Parsimonious and minimally-calibrated models are important to HMC so that physical relationships between regional watershed variables and highly uncertain model parameters might be established. Rather than using tradition inductive measures of streamflow to assess hydrologic similarity for classification, HMC quantifies the hydrologic similarity between two sites as the reciprocating model accuracy, or accuracy of each model when calibrated parameters from the other model are donated to it, and vice versa. Representing hydrologic similarity with model errors produced by a regional range of parameters is a new idea in regionalization that can be used to quantify and reduce parameter uncertainty. Calibrated parameters typically have greater uncertainty than directly calculated parameters. Calibrated parameters are often difficult to define physically and frequently lack data needed for their direct calculation. HMC uses jackknife resampling of complete calibrated parameter sets for all models across the region to generate a model-error matrix of hydrologic similarity spanning the region. The regional error matrix can be interpreted as quantitatively describing parameter uncertainty for the most uncertain parameters across a region. In HMC, the error matrix is used as an inductive basis of hydrologic similarity and combined with a deductive approach to produce a new combined classification that directly incorporates regionalization and reduces parameter uncertainty in models

of ungaged basins. Ultimately, classifying models with reciprocally low errors provides a subset of parameters from a calibrated regional catalog with reduced uncertainty. Figure 2 provides an example overview of the process for HMC with four models.

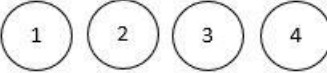

1) Create and calibrate a regional catalog of rainfall-runoff models.

2) Jackknife resample calibrated parameters to generate an error matrix.

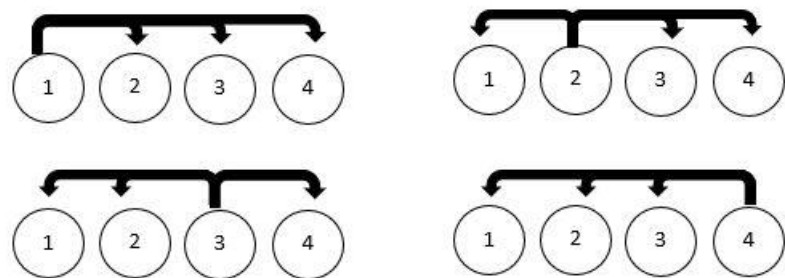

| Model 1 calibration error | Model 2 error with Model 1 parameters | Model 3 error with Model 1 parameters | Model 4 error with Model 1 parameters |
|---|---|---|---|
| Model 1 error with Model 2 parameters | Model 2 calibration error | Model 3 error with Model 2 parameters | Model 4 error with Model 2 parameters |
| Model 1 error with Model 3 parameters | Model 2 error with Model 3 parameters | Model 3 calibration error | Model 4 error with Model 3 parameters |
| Model 1 error with Model 4 parameters | Model 2 error with Model 4 parameters | Model 3 error with Model 4 parameters | Model 4 calibration error |

3) Cluster Analysis on error matrix to group models that estimate streamflow relatively accurately among each other.

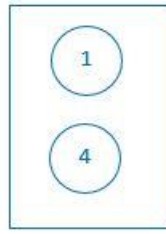 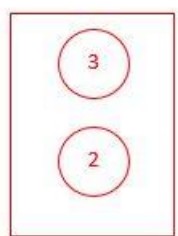

4) Multinomial logistic regression to identify critical landscape characteristics separating clusters

$$Cluster = \beta_1(Drainage\ Area) \times \beta_2(Mean\ Annual Precipitation)$$

**Figure 2: Flowchart overviewing novel Hydrologic Model-based Classification (HMC) using an example with four models.**

### 2.3.1 Hydrologic Models

Hydrological models were created in the US Army Corps of Engineers Hydrologic Engineering Center's Hydrologic Modeling System (HEC-HMS) 4.1 at the 25 gages. Continuous simulations were performed on an hourly time step over WY

2005-2007 to capture a period spanning a wide range of typical hydrologic conditions (WRCC, 2015). Hourly precipitation data were input from the California Irrigation Management Information System (CIMIS), California Data Exchange Center (CDEC), Climate Data Online from the National Oceanic and Atmospheric Administration (NOAA), San Diego County Flood Control District (SDCFCD) and Ventura County Watershed Protection District (VCWPD). CIMIS gages also provided monthly average evapotranspiration data. Independent watershed delineations in ArcMap 10.1 using a 30m digital elevation model from The National Map (USGS, 2019), NHDPlus V2 (McKay et al., 2012), and National Land Cover Database (NLCD) (Fry et al., 2011) were verified by USGS StreamStats data (USGS, 2019). Inverse distance was used to weight precipitation gages from each watershed's centroid. Simple canopy (interception and transpiration) and surface (infiltration) parameters were estimated from delineated data. HEC-HMS model parameters associated with the deficit and constant loss element (infiltration) were calculated directly using soil and imperviousness data available from USGS GAGES-II (Falcone, 2011). Similarly, the time of concentration and Clark unit hydrograph storage coefficient used within the Clark unit hydrograph transform element were calculated directly using the Kirpich method (Kirpich, 1940) and standard approaches utilized by the Arizona Department of Transportation (ADOT, 2014). To produce minimally calibrated models, methods were selected to balance simplicity and parameter parsimony with reliable and process-based hydrology. The Kirpich Method, for example, contains only two parameters, which facilitates straightforward calculations in data-scare areas. It is a long-trusted method for estimating time of concentration (USDA NRCS, 2007) that is highly effective across a wide range of conditions in a similar region (Roussel et al., 2005).

After directly estimating and calculating parameters associated with precipitation losses and hydrograph transformation, only two linear reservoir baseflow parameters were calibrated for the 25 modeled watersheds. Initial flow values were known using streamflow gage data, and a single linear reservoir was used for each of the two groundwater layers. These two layers were connected in parallel with both groundwater layers combining to produce a total baseflow (USACE, 2000). As such, only the groundwater storage coefficient for each layer was altered during calibration.

Flashy floods and periods of little precipitation have strongly influenced the evolution of healthy freshwater aquatic ecosystems in So. CA (Gasith and Resh, 1999). In continuing with this study's focus on streamflow flashiness and permanence as ecologically-relevant management metrics, models were calibrated to optimize RBI and < 1 cfs. While the accuracy of a singular measure of overall fit is typically used for hydrologic model calibration (Bardossy, 2007; Beven, 2012), environmental flow studies have shown it is not ideal for modeling ecological flow metrics (Cassin et al., 2005; Murphy et al., 2013; Parker et al., 2019; Vis et al., 2015). As a result, calibration accuracy of flashiness and flow permanence were equally considered and combined into one "Ecologically-Focused Combined Calibration" (EFCC), which has been used to calibrate hydrologic models for ecological applications in So. CA (Parker et al., 2019). EFCC (Eq. 4) equally weights the percent error (Eq. 3) of RBI (Eq. 1) and < 1 cfs (Eq. 2).

$$Percent\ Error\ (\%) = \left(\frac{|Gage\ flow\ metric - Modeled\ flow\ metric|}{Gage\ flow\ metric}\right) * 100 \qquad (3)$$

$$EFCC\ (\%) = \left[\frac{\left(\frac{|Gage\ RBI - Modeled\ RBI|}{|Gage\ RBI|}\right)*100 + \left(\frac{|Gage<1\ cfs - Modeled<1\ cfs|}{|Gage<1\ cfs|}\right)*100}{2}\right] \tag{4}$$

### 2.3.2 Jackknife Resampling Error Matrix

To compute hydrologic similarity among the regional network of minimally calibrated hydrologic models, storage coefficients and initial discharges of both groundwater layers were donated from one model to all 24 remaining models. This was done for every model in the region in a process known as jackknife resampling (Efron, 1982; Friedl and Stomper, 2014). Model parameters directly calculated or estimated from available landscape data were not jackknifed. Initial baseflow discharges were included in the jackknife analysis and are treated as calibrated parameters because they would be unknown in

a PUB analysis. For each individual model's calibrated parameters, jackknife resampling generated 24 time series characterizing streamflow across the region. The accuracy of each simulated hydrograph resulting from jackknifed parameters was assessed by comparing to the 24 observed USGS streamflow gages. The true gage streamflow data do not affect the jackknifing process because they are only used to determine the accuracy of the output flow data resulting from the jackknifed parameters. The accuracy of each jackknifed parameterization was calculated for the entire 25x24 matrix of time series data

using the EFCC (Eq. 4) scaled by minimum and maximum errors, resulting in a normalized 25x24 matrix quantifying the accuracy of each calibrated model when its calibrated parameters were directly input into all other models. Each sites' original calibration error was added to the matrix such that a normalized 25x25 matrix was produced with very small calibration errors spanning the diagonal.

### 2.3.3 Combined Inductive and Deductive Approach

Combining inductive and deductive approaches for Hydrologic Model-based Classification was very similar to the combined approach under traditional classification that implemented multinomial logistic regression. Using the jackknife error matrix of hydrologic similarity, weighted classical (metric) multidimensional scaling, PCA, and a scree plot provided a sense of how sites might cluster. K-means clustering with C-Index, Dunn, McClain, and Silhouette indices was used to split sites into reciprocating low model-error clusters. This inductive approach produced groups of hydrologically similar gages, as

measured by a site's ability to accurately model all other sites within its group. A deductive approach was added to HMC by using multinomial logistic regression to determine if watershed variables could predict low-error cluster membership.

### 2.4 Classification Assessment

    To better understand the utility of each classification towards estimating flow in ungaged basins, a performance metric dubbed "average cluster error" (ACE) was developed for this study. ACE characterizes the errors produced by donated

parameters within a classification method and its classes. Low-error classifications and classes indicate greater certainty in donated calibrated parameters, which inherently contain high uncertainty in models of ungaged basins. Classifications and classes with low ACE values may provide the foundation for accurately modeling ungaged basins with regionalization. ACE

was modeled after the cross-validation standard error (CVSE) statistic presented by Wortman (2005) and is displayed in Eq. 5, wherein C is the total number of clusters produced by a specific classification, c represents each cluster, S is the total number of sites within the given cluster, s is each site from the cluster, Normalized Errors is taken directly from the jackknife error matrix, and P is the total number of sites (25 in this study).

$$Average\ Cluster\ Error = \frac{\sum_{c=1}^{C}\sum_{s=1}^{S}(Normalized\ Errors_s)}{P} \tag{5}$$

The following example helps explain how Eq. 5 was used: Say a specific classification divided the 25 sites into 5 equal groups split chronologically (Sites 1-5, 6-10, 11-15, etc.). Total error for the first group would be computed by summing all within cluster errors (when site 1 parameters were applied to Sites 2, 3, 4, and 5; when site 2 parameters were applied to Sites 1, 3, 4, and 5; etc. for site 3, 4, and 5 parameters). This same process would be repeated for the four remaining groups and summed to produce a final total error. The total error would be divided by 25 sites to yield a single metric quantifying the average model error across all sites, exclusive to a specific classification. Following this procedure, ACE values can also be computed for individual clusters unique to one classification, wherein the number of sites assigned to the specific group of interest would take the place of P (P = 5 when only considering one cluster from the example above), and the $\sum_{c=1}^{C}()$ term would not be used because only one cluster from the classification is considered. Because all sites receiving each model's parameters were treated as ungaged basins during jackknife resampling, the ACE statistic provides insight regarding how well different classifications, or different groups within one classification, might be incorporated into regionalization.

Additionally, the adjusted Rand index (ARI) was computed between each traditional classification technique and Hydrologic Model-based Classification to compare the similarity of any two unique classification. ARI typically ranges from 0 to 1, wherein a value of 0 indicates no similarities between clusters and a value of 1 represents identical clusters; however, negative values can occur if class similarity is less than what would be expected during random clustering (Hubert and Arabie, 1985). Essentially, ARI values near 0 indicate a classification scheme provides unique groups that do not overlap. Specifically, the "clues" package in R (Chang et al., 2010) was implemented to compute an ARI between all suitable classifications.

Between the two measures for assessing classifications in this study, ARI provides an understanding of each classification's ability separate its data, while ACE reflects the ability of a classification, or cluster within a classification, to estimate streamflow in ungaged basins. ARI is a more general metric for insight into data clustering, while ACE is a specific metric focused on cluster performance in ungaged basins. More generally, ARI quantifies between cluster variability while ACE quantifies within cluster variability.

 **3 Results**

**3.1 Traditional Classification**

**3.1.1 Inductive Approach**

Classification of hourly flashiness and flow permanence metrics in coastal southern CA resulted in three classes (Fig. 3). Sites were essentially split according to flow permanence with intermittent streams containing below-average flashiness (Class 1 with 6 sites), perennial streams spanning the full range of flashiness (Class 2 with 10 sites), and ephemeral streams spanning the full range of flashiness (Class 3 with 9 sites). The intermittent class contained the smallest average cluster error with the least within cluster variability (0.2, Fig. 3), indicating calibrated parameters from models of these streams possessed the least uncertainty. Likewise, the perennial class had the least utility towards ungaged basins because it contained the most within cluster variability (ACE = 0.9, Fig. 3). When considering all three clusters produced by traditional inductive classification, the ACE was 0.6 (Fig. 3).

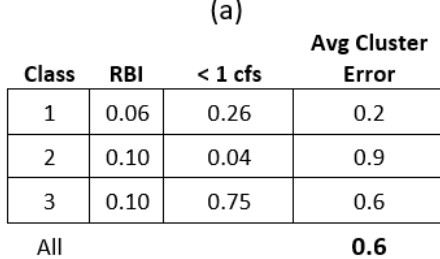

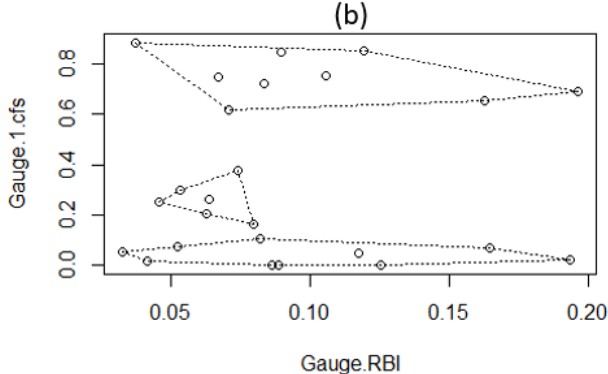

| Class | RBI | < 1 cfs | Avg Cluster Error |
|-------|------|---------|-------------------|
| 1 | 0.06 | 0.26 | 0.2 |
| 2 | 0.10 | 0.04 | 0.9 |
| 3 | 0.10 | 0.75 | 0.6 |
| All | | | **0.6** |

**Figure 3: Results of inductive approach to traditional classification. Specifically, (a) mean predictor metric values and ACE for the different classes and overall classification; (b) ordination plot illustrating metric values across clusters.**

**3.1.2 Deductive Approach**

Classification of watershed characteristics yielded five classes with drainage area and soil content, specifically the percentage of Hydrologic Soil Group C (HGC), providing a parsimonious classification (Fig. 4). These two watershed variables were log-transformed within the K-means algorithm to address the right skewed nature of drainage area caused by a few large basins. Sites were primarily divided by drainage area, and secondarily by HGC, to generate classes of small basins with low HGC (Class 3 with 3 sites), small basins with high HGC (Class 5 with 7 sites), medium-sized basins with low HGC (Class 1 with 5 sites), medium-sized basins with high HGC (Class 2 with 7 sites), and large basins with high HGC (Class 4 with 3 sites). The large basin with high HGC class contained the smallest ACE (0.2, Fig. 4), while the medium-sized basin with low HGC provided the largest (0.6, Fig. 4). An ACE of 0.4 was computed after considering all five clusters produced by traditional deductive classification (Fig. 4).

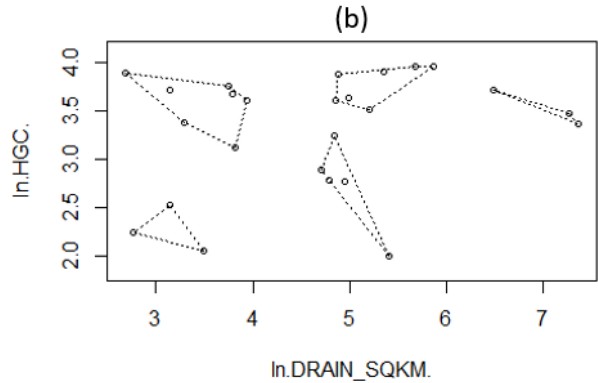

**(a)**

| Class | Drainage Area (km²) | HGC % | Avg Cluster Error |
|-------|---------------------|-------|-------------------|
| 1 | 143.9 | 17 | 0.6 |
| 2 | 206.6 | 44 | 0.5 |
| 3 | 24.0 | 10 | 0.3 |
| 4 | 1220.6 | 34 | 0.2 |
| 5 | 35.4 | 37 | 0.5 |
| All | | | **0.4** |

Figure 4: Results of deductive approach to traditional classification. Specifically, (a) mean predictor metric values and ACE for the different classes and overall classification; (b) ordination plot illustrating metric values across clusters.

### 3.1.3 Combined Inductive and Deductive Approaches

Neither an expanded cluster analysis nor predicting inductively and deductively produced clusters with the selected watershed characteristics and flow metrics, respectively, improved classification over the individual inductive and deductive approaches. New multinomial regression models were developed to accurately predict traditional inductive clusters with drainage area, % clay soil, minimum elevation, and annual minimum precipitation, and predict gage reference status with drainage area, % silt soil, baseflow index, and relative humidity.

### 3.2 Hydrologic Model-based Classification

### 3.2.1 Models

Calibration of the 25 HEC-HMS models at USGS gages produced models with extremely accurate flashiness and flow permanence. Average percent errors of both RBI and < 1 cfs were well under 1%.

### 3.2.2 Combined Inductive and Deductive Approach

Hydrologic Model-based Classification combined inductive and deductive classification to produce a multinomial logistic regression model (deductive classification) that uses landscape variables to predict membership of five hydrologically-similar groups of models (inductive classification) (Table 1). The inductive approach used in HMC does not group sites by the similarity of measured or modeled metrics, as is done traditionally, but instead groups sites to maximize model accuracy when calibrated models' parameters are donated to all other sites within a group. Despite this important distinction, streamflow flashiness and permanence were well distributed across the five hydrologic model-based clusters (Table 1). A multinomial logistic regression model was able to predict low-error class membership with 4% error (24 sites matched correctly) using drainage area, sandy soil content, mean annual precipitation, and mean annual minimum precipitation. The number of sites

was distributed less evenly across classes for Hydrologic Model-based Classification than traditional methods, with the first two clusters containing two sites each, the third cluster containing three, the fourth containing five sites, and the final cluster containing over half the sites with 13. As such, it is no surprise that class five contained the largest within cluster variability (ACE = 0.5, Table 1), and is subsequently its worst performing group in ungaged basins. However, no other class within HMC produced an ACE greater than 0.1, which contributes to HMC owning the lowest within cluster variability across all classifications (ACE = 0.3, Table 1).

Stream classes produced by HMC include medium-sized basins with flashiness on both the high (Class 1) and low (Class 4) end. Flashy Class 1 streams receive the least precipitation and are located in southern San Diego County. Non-flashy Class 4 streams comprise the two eastern-most sites. Medium-small basins (Class 3) receive relatively little precipitation and are located near the coast, while large-medium basins (Class 5) receive the most precipitation and are spread throughout the study area. The largest basins (Class 2) are slightly flashier and drier than the large-medium basins (Class 5). These Class 2 streams are concentrated in the northern area of the study area.

| Class | Drainage Area (km$^2$) | Sand % | Annual Avg Precip (cm) | Annual Min Precip (cm) | RBI | < 1 cfs | Avg Cluster Error |
|-------|------------------------|--------|------------------------|------------------------|------|---------|-------------------|
| 1 | 146.4 | 41 | 35 | 1.9 | 0.16 | 0.44 | 0.1 |
| 2 | 463.8 | 38 | 51 | 1.0 | 0.10 | 0.33 | 0.1 |
| 3 | 93.4 | 33 | 39 | 0.6 | 0.12 | 0.40 | 0.0 |
| 4 | 151.9 | 59 | 40 | 1.6 | 0.05 | 0.58 | 0.1 |
| 5 | 222.5 | 52 | 55 | 1.8 | 0.08 | 0.28 | 0.5 |
| All | | | | | | | **0.3** |

| Logistic Regression Landscape Variable | Definition | Source |
|----------------------------------------|------------|--------|
| DRAIN_SQKM | Total upstream drainage area (km$^2$) | NHDPlus V2 (McKay et al., 2012) |
| SANDAVE | Percentage of sandy soil (%) | GAGES-II (Falcone, 2011) |
| PPTAVG_CAT | Mean annual precipitation of NHD catchment (cm) | GAGES-II (Falcone, 2011) |
| CAT_AnnMinPrecip | Mean annual minimum precipitation of NHD catchment (cm) | GAGES-II (Falcone, 2011) |

**Table 1: Results of Hydrologic Model-based Classification (HMC).**

### 3.3 Adjusted Rand Index (ARI)

The geographical distribution of four unique classifications is displayed in the Appendix (Fig. A1), including traditional inductive (flow metrics), traditional deductive (watershed characteristics), a hybrid inductive/deductive (GAGES-II reference sites), and hydrologic model-based as a hybrid inductive/deductive (model accuracy and watershed characteristics). Results of the ARI analysis show no major similarities and large variability between classifications, with the

strongest relationship between GAGES-II reference sites and inductive classification (ARI = 0.12, Table 2). Inductive and Hydrologic Model-based Classifications were most different with an ARI of -0.04 (Table 2).

|  | Inductive | Deductive | Reference | Hydrologic model-based |
|---|---|---|---|---|
| Inductive | - | -0.01 | 0.12 | -0.04 |
| Deductive | -0.01 | - | 0.004 | 0.09 |
| Reference | 0.12 | 0.004 | - | 0.013 |
| Hydrologic model-based | -0.04 | 0.09 | 0.013 | - |

**Table 2: Adjusted Rand Index (ARI) among four unique classifications. The strongest relationship between classifications is underlined.**

**4 Discussion**

Hydrologic Model-based Classification introduces a new way to think about stream similarity, which can improve the accuracy of hydrologic modeling and environmental flow management in ungaged basins. For hydrologic modeling, HMC can be incorporated into iterative development of a hydrologic foundation, and it supplies the foundation for an improved approach to regionalization of ungaged basins. As a management tool, HMC streamlines priority environmental flow metrics

in ungaged basins.

**4.1 Hydrologic Model-based Classification and environmental flow management**

Using Hydrologic Model-based Classification to incorporate regionalization for modeling ungaged basins into stream classification provides an opportunity to improve environmental streamflow studies that require ungaged data. ELOHA is an iterative process with significant feedback loops; however, stream classification is recommended to occur second, after

developing a hydrologic foundation, and no guidance is provided on how classification might inform the hydrologic foundation or vice versa (Poff et al., 2010). Because the hydrologic foundation generates baseline and current hydrographs at sites with bioassessment data, many of which are ungaged, reciprocally low-error classes produced by HMC could be utilized in a modeling framework to increase the hydrologic foundation's accuracy. Switching the order of the first two steps in ELOHA, and first classifying sites using HMC, could improve streamflow estimation in ungaged basins as a part of the hydrologic

foundation. At the very least, developing the hydrologic foundation could be iterative with classification as key characteristics of the sites become better understood, especially if ungaged basins must be modeled.

The primary role of stream classification, as one of the four major steps of ELOHA, is to strengthen and standardize regional flow-ecology relationships so that they may be better implemented for water management (Poff et al., 2010); however, it is the one step of ELOHA some studies have determined unnecessary and bypassed (Kendy et al., 2012). To this point, large-

scale classifications in the Chesapeake Bay watershed (Buchanan et al., 2011) and Western US, including a separate classification in California, (Hawkins and Vinson, 2000) did not improve benthic macroinvertebrate explanatory power. While this study has demonstrated how the primary application of stream classification is useful in coastal southern California, it has also introduced HMC to extend classification beyond its traditional role to modeling ungaged basins for developing a hydrologic foundation in any region. Not only would a more accurate hydrologic foundation create more accurate flow-ecology

relationships and stronger environmental flow criteria, but it would also improve the utility of stream classification within ELOHA.

Modeled streamflow data does not always classify streams the same as gage data for the same sites. Peñas et al. (2016) showed daily and monthly gage data clustered better than monthly modeled data in Spain. Similarly, modeled data provided different classes than gaged data in North Carolina (Eddy et al., 2017). While model accuracy is always a high priority in

hydrologic applications, stream classification is very sensitive to this accuracy, which underscores the importance of accurate models within ELOHA. Poor model accuracy not only directly diminishes the utility of flow-ecology relationships, and subsequent environmental flow recommendations, but it can indirectly hamper management efforts by providing inconsistent stream classes. When ungaged basins are considered in ELOHA, model accuracy must be highly prioritized or else lingering and compounding errors might spoil otherwise legitimate efforts.

From an operational perspective, Hydrologic Model-based Classification is more time-consuming than traditional classifications and might become unwieldy when applied across an expansive geographic region with many sites to classify. This is because not only must hydrologic models be created and calibrated for every classified site, but each model must be analyzed with every other model's calibrated parameters to produce the critical jackknife resampling error matrix. If ungaged basins are to be included, however, some extra time spent building models is recouped as they would have been built anyway

under traditional classifications. This study has demonstrated that HMC is feasible for 25 sites spanning a fairly large and highly heterogeneous region in the south coast of California. If a significantly larger region or denser network was the focus of this study, HMC would likely provide even more precise classes and accurate streamflow estimates, but with a substantially greater time investment. Realistically, HMC becomes less feasible at a state-wide scale or for a large network (~50 sites). HMC uses conceptual hydrologic models with process-based methods, which can be created and calibrated relatively quickly,

but with uncertainty (Knoben et al., 2019). These issues make HMC most effective for moderate-scale environmental flow development, which might range from basin-level to spanning multiple counties, or with expeditious hydrologic models. While HMC is more time-consuming than traditional classifications, it was developed with simple, lumped hydrologic models and time-invariant parameters. Other sophisticated modeling approaches have been developed with more complicated model structures, such as adaptive clustering with distributed models (Ehret et al., 2020) and diagnostic evaluations with time-variable

parameters (de Vos et al., 2010).

## 4.2 Stream classification for regionalizing ungaged basins

Hydrologic Model-based Classification not only provides new information characterizing regional streams complementary to traditional classifications, but it can also be used to accurately model ungaged basins across heterogenous area through regionalization, as evidenced through the average cluster error metric describing within cluster variability. ACE unpacks important information buried inside the jackknife resampling matrix describing how accurately a set of calibrated parameters can be donated from its original model to all other models in the region, as if the other models were ungaged. Error values from the matrix can be assessed for each model in the region or, when performing stream classification, can be aggregated to quantify ACE for every class within a given classification. Further aggregation can provide an overall measure of ungaged modeling accuracy for an entire classification approach to compare to other classification schemes. A comparison of these overall ACE values shows Hydrologic Model-based Classification containing the least within cluster variability, which provides the most certainty regarding parameters in models of ungaged basins (ACE 0.3; Table 1). HMC was followed by deductive classification with drainage area and HGC (ACE 0.4; Fig. 4), inductive classification with < 1 cfs and RBI (ACE 0.6; Fig. 3), and lastly GAGES-II reference status (ACE 1.4).

By providing a method for reducing parameter uncertainty in models of ungaged basins, HMC has demonstrated utility beyond complementary classification. Modeling ungaged basins is fundamental to ELOHA (Poff et al., 2010) and many other hydrology applications, but different approaches vary significantly, contain uncertainty, and do not perform particularly well across a geologically and hydroclimatically diverse area (Arsenault et al., 2019; Blöschl et al., 2013). This study provides a foundation for directly incorporating the regional accuracy of a catalog of hydrologic models into a framework for improving ungaged modeling within a heterogeneous region.

The five traditional classes with low measures of ACE (≤ 0.5) (Fig. 3 and Fig. 4) provide additional information to reduce ungaged model uncertainty in So. CA. This study has shown flow permanence and flashiness were more consistently modeled in ungaged basins containing intermittent streams than ephemeral or perennial streams. Extreme sensitivity to precipitation explains why ephemeral streams did not produce a low ACE, and, while initially, it may be surprising to see baseflow parameters more accurately interchanged between models of intermittent streams than perennial, the effluent nature of perennial streams, especially in a region as rapidly urbanizing as So. CA, inconsistently augments the natural flow regime (Ponce and Lindquist, 1990), and likely prevented accurate modeling in this study. Similarly, flows were modeled with more certainty at GAGES-II reference sites (ACE 0.4) than non-reference (ACE 1.9), wherein flow alteration restricts the ability to transform precipitation into streamflow. Based on the results of this study, intermittent reference streams are likely most accurately regionalized in the south coast.

While no combined classification in coastal southern CA was able to predict class membership of all 25 sites with 100% accuracy, HMC came the closest. This finding underscores the potential for using a measure of model accuracy across a region to define hydrologic similarity within stream classification. Olden et al. (2012) split deductive classification into three sub-approaches: "environmental regionalization" to provide a spatial representation of stream similarity, "hydrologic

regionalization" using models to estimate flow in ungaged basins, and "environmental classification" for geographically independent classification; however, only one inductive approach, ideal for geographic independence, is described: "streamflow classification". The new Hydrologic Model-based Classification developed in this study is based on inductive reasoning but is not "streamflow classification". Instead HMC is a type of "streamflow regionalization" wherein each region is a reciprocally low-error class. Instead of defining geographic areas of assumed flow similarity using watershed characteristics, "streamflow regionalization" directly groups sites based on modeled flow similarity. This new approach essentially hybridizes "hydrologic regionalization" and "streamflow classification".

Deductive classification produced relatively low uncertainty of model parameters, with all five classes containing ACE values between 0.2 and 0.6 (Fig. 4). The relatively tight spread coupled with a low overall ACE (0.4; Fig. 4) implicate deductive classification as a worthy alternative to HMC for regionalization of ungaged basins. These results are consistent with the most common implementation of regionalization wherein models are typically grouped by spatial proximity, physical similarity, or parameter regression (Oudin et al., 2008; Razavi and Coulibaly, 2013; Samuel et al., 2011). This study has shown how a new type of "streamflow regionalization", akin to Hydrologic Model-based Classification, might edge out traditional "hydrologic regionalization" from deductive classification, at estimating streamflow in ungaged basins. "Hydrologic regionalization" and "streamflow regionalization" both implement watershed characteristics to separate sites for high utility in modeling ungaged basins; however, "streamflow regionalization" improves modeling by directly incorporating a quantifiable measure of ungaged model accuracy. This important addition to "streamflow regionalization" directly captures regional model uncertainty and strengthens the science supporting modeling ungaged basins.

**4.3 Stream classification in coastal southern California**

Each of the different classifications described in this study provides unique information on how coastal southern CA streams might be stratified for environmental flow management. Previous state-wide classifications by Lane et al. (2017), Pyne et al. (2017), and Lane et al. (2018) are too broad to provide the resolution needed for sub-daily hydrologic modeling in the South Coast subregion, which is characterized by heterogeneous land use and geologic settings. While this study was limited to 25 sites in So. CA, flow permanence is clearly an important metric for grouping streams, as demonstrated in the inductive approach, while drainage area and the percentage of relatively low infiltration high runoff soils proved most important for deductively classifying streams. The slightly negative ARI between inductive and deductive classifications (-0.01, Table 2) indicates no similarity between the classes produced by the different approaches (Hubert and Arabie, 1985). These highly different classifications provide complementary information in the south coast and suggest a relationship exists between flow permanence, drainage area, and HGC. This weak relationship was identified as, at best, just under half (12/25) of inductively and deductively produced clusters, could be correctly predicted using multinomial logistic regression. While this level of accuracy is not acceptable for practical stream classification, it does establish a non-random relationship between flow permanence, drainage area, and HGC in the study region.

As measured by ARI, traditional inductive classification and reference status classification were the two most similar, but still contained high variability (0.12, Table 2). This finding is consistent with how GAGES-II primarily uses flow alteration to classify reference streams (Falcone, 2011), and with how ELOHA recommends classifying by hydrologic similarity to develop flow-ecology relationships (Poff et al., 2010). Furthermore, the reference status classification established a relationship, predominately with drainage area, but also silt content, baseflow index, and relative humidity, which could help water managers identify streams facing potential flow alteration.

The two most different classifications in this study were traditional inductive and hydrologic model-based (ARI - 0.04, Table 2). Hydrologic Model-based Classification is primarily based on an inductive approach; however, it quantifies hydrologic similarity completely differently than traditional inductive classification. The negative non-random relationship between these classifications is explained as the traditional approach considers gage data similarity and Hydrologic Model-based considers model data similarity of the same metrics. The differences in these two inductively-based classifications underscore the complexity in modeling streamflow permanence and flashiness in So. CA and suggest great effort must be taken when modeling ungaged basins in the south coast region.

Using ARI, this study has demonstrated how four unique stream classifications can each provide important, complementary information regarding how streams across a region may be grouped for management. While the two inductively-based classifications appear the most useful for separating gaged and ungaged sites, respectively, important relationships and management opportunities can be revealed through a robust regional stream classification using multiple approaches.

## 5 Conclusions

Accurately modeling ungaged basins is often necessary for quantification and management of environmental streamflows (Poff et al., 2010), but it is a difficult undertaking with no consensus approach among the hydrology community, especially in heterogenous regions (Arsenault et al., 2019; Blöschl et al., 2013). Furthermore, stream classification is one of the four major steps used to develop environmental flow criteria within ELOHA (Poff et al., 2010), but it is not always used in the framework (Kendy et al., 2012). This study sought to increase the utility of classification within ELOHA while simultaneously strengthening the science supporting modeling and management of ungaged basins in heterogeneous regions. To this end, Hydrologic Model-based Classification was developed to provide: complementary classification information, improved ungaged model accuracy, and new opportunities for stream management. Iterating between the first two steps of ELOHA (hydrologic foundation and classification) within HMC improves both steps and produces stronger environmental flow criteria.

While this study focused on streamflow permanence and flashiness due to their known ecological importance in the study region (Gasith and Resh, 1999; Mazor et al., 2018; Parker et al., 2019), additional flow metrics corresponding to other element of the flow regime are ecologically-relevant in So. CA (Yarnell et al., 2020) and could be incorporated. To develop a

better understanding of HMC in general, it could be extended to new regions and compared to the results of this study. This could produce general relationships between different classifications and provide insight into which classification approach might be most appropriate for specific applications and regions. Likewise, a type of nested classification similarly implemented across many regions would help different stakeholders understand how management actions at multiple geographic scales might affect streams and would foster coordinated management relationships. As HMC is expanded to additional regions, a better understanding of the similarity of within-class management plans will be developed. These findings will be highly dependent on the management metrics and regions, but a general sense for management plan transferability within low-error classes will offer a clearer understanding of how Hydrologic Model-based Classification might assist in ungaged stream management without ever modeling the basin.

For coastal southern California, HMC results from this study should be further developed into a full framework for modeling time-series of discharge in new ungaged basin(s) from the heterogenous region. This would foster a better understanding of the modeling complexities within Hydrologic Model-based Classification, and its associated new regionalization framework, and would provide the basis of a hydrologic foundation prioritizing ungaged basins, which is needed to develop robust regional environmental flow criteria in So. CA.

## 6 Acknowledgements

We would like to thank our technical advisory and stakeholder workgroups for their continued participation throughout this project. Their input improved the technical quality and management applicability of this study. Support for this project was provided by the California State Water Resources Control Board. The contents of this document do not necessarily reflect the views and policies of the State Water Resources Control Board, nor does mention of trade names or commercial products constitute endorsement or recommendation for use.

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

**8 Appendix**

| Site Name | USGS Gage | Reference | Area (km²) | Impervious % | HGC | Clay % | Silt % | Sand % | Basin Min Elevation (m) | Catchment Annual Min Precip (cm) | Catchment Annual Avg Precip (cm) | BFI % | RH % | Mean Annual Flowrate (cfs) | | |
|---|---|---|---|---|---|---|---|---|---|---|---|---|---|---|---|---|
| | | | | | | | | | | | | | | WY 2005 | WY 2006 | WY 2007 |
| Andreas | 10259000 | Ref | 23.2 | 0.0 | 12 | 10 | 30 | 59 | 177 | 1.0 | 32 | 49 | 42 | 6.1 | 2.2 | 0.9 |
| Arroyo Seco | 11098000 | Ref | 42.5 | 0.5 | 43 | 18 | 48 | 34 | 398 | 1.0 | 63 | 38 | 54 | 52 | 8.6 | 0.9 |
| Arroyo Trabuco | 11047300 | Non-ref | 141.4 | 19.9 | 16 | 25 | 43 | 33 | 18 | 1.0 | 34 | 24 | 62 | 69 | 13 | 5.1 |
| Campo | 11012500 | Non-ref | 222.3 | 7.0 | 7.4 | 10 | 20 | 70 | 623 | 2.0 | 42 | 41 | 47 | 2.0 | 0.4 | 0.1 |
| Carpinteria | 11119500 | Non-ref | 45.4 | 0.1 | 23 | 23 | 42 | 34 | 6.6 | 0.0 | 44 | 30 | 54 | 18 | 3.7 | 0.0 |
| Deep Creek | 10260500 | Non-ref | 354.0 | 2.4 | 52 | 12 | 24 | 64 | 915 | 2.0 | 23 | 33 | 40 | 171 | 63 | 7.8 |
| Devil Canyon | 11063680 | Non-ref | 14.7 | 0.7 | 49 | 14 | 32 | 54 | 598 | 3.0 | 80 | 40 | 47 | 2.7 | 4.9 | 2.1 |
| East Twin | 11058500 | Non-ref | 23.1 | 0.7 | 41 | 14 | 31 | 56 | 483 | 2.2 | 61 | 34 | 45 | 2.7 | 5.4 | 1.6 |
| Jamul | 11014000 | Non-ref | 182.9 | 0.5 | 33 | 19 | 39 | 43 | 153 | 2.0 | 36 | 34 | 56 | 23 | 0.1 | 0.0 |
| Lytle | 11062000 | Non-ref | 119.8 | 0.4 | 16 | 11 | 31 | 58 | 725 | 2.0 | 91 | 54 | 48 | 37 | 32 | 3.1 |
| Matilija | 11114495 | Ref | 128.5 | 0.0 | 37 | 22 | 42 | 36 | 348 | 0.0 | 72 | 39 | 50 | 156 | 37 | 4.3 |
| Plunge | 11055500 | Non-ref | 44.2 | 1.3 | 40 | 13 | 31 | 55 | 485 | 2.4 | 45 | 29 | 43 | 14 | 7.9 | 2.1 |
| Poway | 11023340 | Non-ref | 110.0 | 21.8 | 18 | 23 | 38 | 39 | 75 | 1.0 | 34 | 24 | 60 | 36 | 7.3 | 4.5 |
| Rainbow | 11044250 | Non-ref | 27.0 | 4.3 | 29 | 15 | 31 | 54 | 161 | 1.0 | 45 | 30 | 59 | 16 | 1.4 | 0.4 |
| San Luis Rey | 11042000 | Non-ref | 1433.8 | 3.1 | 32 | 15 | 28 | 57 | 1.2 | 1.0 | 31 | 35 | 53 | 229 | 29 | 9.7 |
| San Mateo | 11046300 | Ref | 210.2 | 0.1 | 50 | 17 | 35 | 47 | 120 | 1.7 | 46 | 26 | 56 | 90 | 3.3 | 0.1 |
| Sandia | 11044350 | Non-ref | 51.1 | 1.3 | 37 | 19 | 39 | 42 | 124 | 1.0 | 45 | 28 | 59 | 30 | 6.1 | 4.0 |
| San Jose | 11120500 | Ref | 15.8 | 0.4 | 9.5 | 20 | 45 | 35 | 26 | 0.7 | 48 | 19 | 61 | 14 | 3.0 | 0.2 |
| Santa Margarita Sump | 11044300 | Non-ref | 1576.9 | 4.6 | 29 | 15 | 30 | 55 | 99 | 1.0 | 43 | 29 | 51 | 82 | 18 | 9.4 |
| Santa Ysabel | 11025500 | Non-ref | 290.9 | 0.1 | 52 | 16 | 31 | 52 | 247 | 2.0 | 43 | 32 | 49 | 29 | 1.7 | 0.0 |
| Santa Maria | 11028500 | Non-ref | 147.7 | 2.6 | 38 | 17 | 32 | 51 | 397 | 2.0 | 44 | 27 | 53 | 16 | 0.4 | 0.1 |
| Santiago | 11075800 | Non-ref | 32.9 | 0.1 | 7.8 | 23 | 46 | 31 | 374 | 1.0 | 49 | 25 | 59 | 20 | 1.8 | 0.0 |
| Sespe Fillmore | 11113000 | Non-ref | 651.0 | 0.1 | 41 | 22 | 42 | 36 | 168 | 0.0 | 52 | 45 | 45 | 515 | 211 | 15 |
| Sespe Wheeler Springs | 11111500 | Ref | 131.9 | 0.1 | 48 | 22 | 41 | 37 | 1028 | 0.0 | 69 | 40 | 45 | 87 | 22 | 1.4 |
| Sweetwater Descanso | 11015000 | Ref | 126.0 | 0.3 | 26 | 13 | 25 | 62 | 754 | 3.0 | 62 | 40 | 46 | 21 | 3.7 | 1.0 |

**Table A1: Study sites details. Drainage area data are from the National Hydrography Dataset Plus Version 2 (NHDPlus V2) (McKay et al., 2012), impervious data are from StreamStats (USGS, 2019), and all other watershed characteristics, including reference gage status, are from GAGES-II (Falcone, 2011). Mean Annual Flowrates were computed from gage data.**

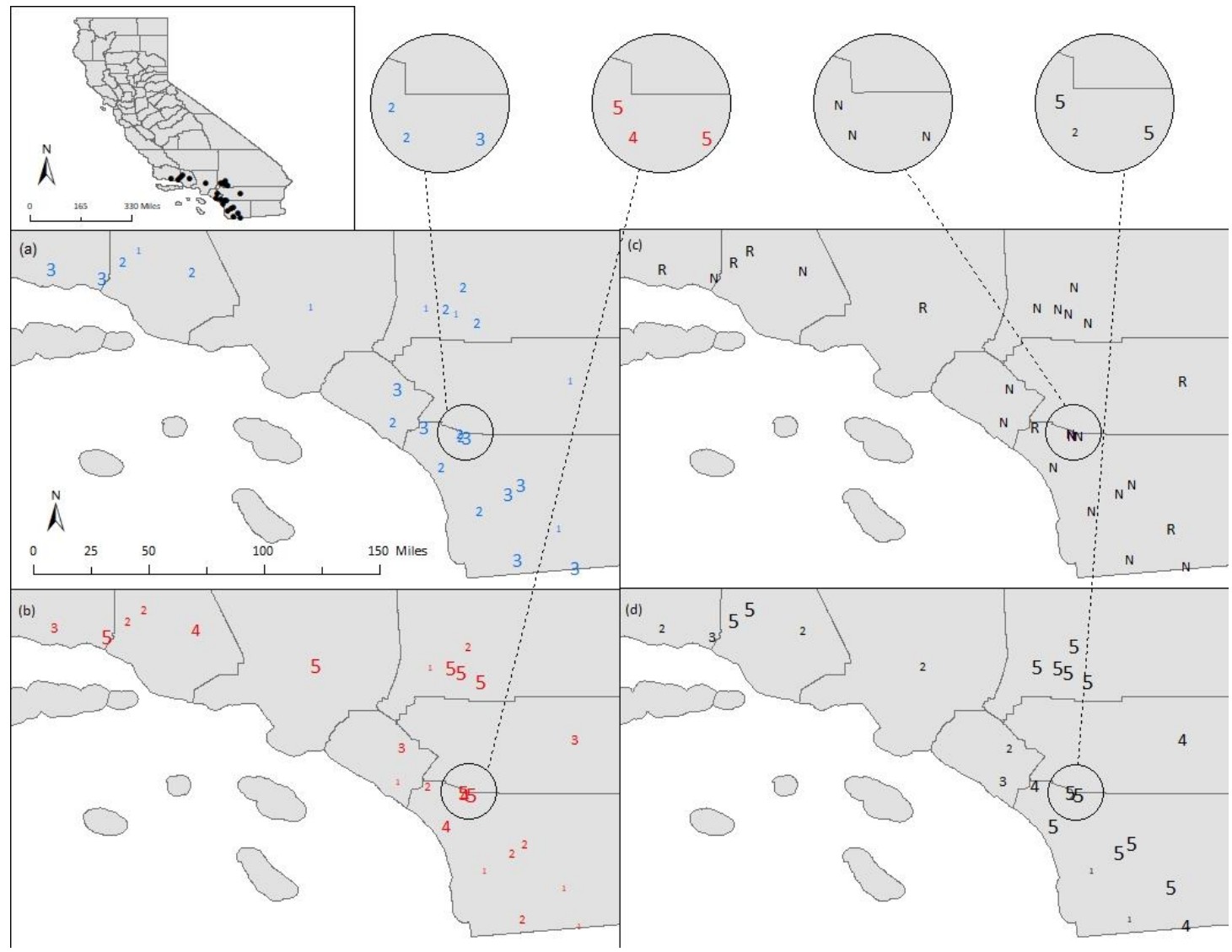

**Figure A1: Geographical distribution of classes, specifically for (a) traditional inductive approach; (b) traditional deductive approach; (c) GAGES-II reference sites; (d) new hydrologic model-based method.**