# Peer review of "Advancing stream classification and hydrologic modeling of ungaged basins for environmental flow management in coastal southern California"

_Hydrology and Earth System Sciences, 2021_

## Author Response (AR1)

HESS Editorial Team,

Thank you to the reviewers for the excellent feedback and opportunity to improve our manuscript. We have accepted and responded to all referee comments (RC1 and RC2).

In regard to RC1's general feedback, we have bolstered the background with additional literature on hydrology and stream classification of flashy and ephemeral streams (revised revised lines 36-38; 100-101). We have also included additional language highlighting flashy and ephemeral streams (revised lines 16-17; 18; 45). We more clearly articulated differences between modeling general hydrology and modeling environmental flows (revised lines 81-83). We clarify "best current practices" (revised line 105). Additionally, we strengthened the background and discussion related to hydrologic model error and uncertainty (revised lines 196-198; 415-416).

For specific RC1 comments, we have corrected our interpretation of Lane et al. (2017) (revised lines 60-61). We changed "separate" to "separation" (revised line 65). We defined all acronyms prior to using (revised line 39, 45, 58, etc.). We substantially changed Figure 1 to eliminate county delineations, added the cities of Santa Barbara, Los Angeles, and San Diego, and changed the inset to show the western US. We specified "daily average streamflow" (revised line 161). We rewrote the final paragraph in Section 2.2.1 to provide more detail about methods and scripts for stream classification (revised lines 165-171). We also removed mention of specific R packages from the main text. The following paragraph in Section 2.2.2 was rewritten to provide further discussion of how highly correlated metrics were removed and to remove mention of specific R packages (revised lines 175-178). We removed mention of specific R packages from the following paragraph in Section 2.2.3 (revised line 183). Figure 2 was completely redeveloped into a more intuitive flowchart to provide a better visual tool for understanding this complex process. Finally, we reworded the reporting of our results to clarify model calibration was accurate, not "successful" (revised lines 336-337).

We addressed RC2's general feedback by augmenting our background literature related to ungaged catchments (revised lines 91-93; 94-96). For specific RC2 comments, we clarified "singular streamflow metrics" (revised line 76) and "network of models" (revised lines 85-86). We added transferring hydrologic index (revised line 87). We specified that we used "conceptual" models and not just process-based (revised line 76-77; 415-416). We ensured all equations use proper mathematical notation required by HESS. We provided more detail on why we chose 1 cfs as the threshold for a dry stream (revised line 155). We enhanced our description of "weighted classical (metric) multidimensional scaling (revised lines 166-167) and clustering indices (revised lines 168-169). We clarified our statement that "calibrated parameters inherently have greater uncertainty than directly calculated parameters" (revised lines 196-198). We also clarified "reciprocating model accuracy" (revised lines 193-194). We completely redeveloped Figure 2, which was also requested by RC1. We clarified "successful" model calibration (revised lines 336-337), which was also requested by RC1. Finally, we added DOIs to citations.

---

## Author Response (AR3)

Thank you to the Editor for accepting our paper for publication. We have uploaded the Production File and have made one final change:

**Remarks from the preceding review file validation:**

***It seems that table is included as figure #5. If it is so, it must be re-labelled as table and the references in the manuscript text must be adjusted accordingly.***

We have changed "Figure 5" to "Table 1". We have changed our original "Table 1" to "Table 2". We have corrected all references in the manuscript to reflect the final "Table 1" and "Table 2".

From previous submission:

Thank you to the Editor and Reviewers for their excellent feedback and the opportunity to improve our manuscript. We greatly appreciate your help. We found all comments useful and constructive. As such, we have accepted and responded to all Referee comments (RC1 and RC2) and Editor comments. Below, we reproduce each Referee and Editor comment and describe the changes made in our manuscript. In the description of our revisions below, all line numbers refer to the revised manuscript.

**Editor general comments:**

***I am returning to the authors for another round of minor revisions. The revisions as proposed were very minimal and did not substantially engage with the reviewer comments, particularly the overview comments of Reviewer 1. These comments ask for improved background and literature for several topics, each of which should at the minimum include a new paragraph in the paper with multiple references.***

We have endeavored to substantially engage with the reviewer comments and provide thorough and robust responses, especially in regard to the general comments provided by Reviewer 1. We have added a new paragraph at the beginning of the Introduction focused on physical hydrologic processes of non-perennial streams (Lines 37-48). We cited 6 new references (Skoulikidis et al., 2017; Stubbington et al., 2018; Datry et al., 2014; Tooth, 2000; Gannon et al., 2022; Hammond et. al, 2020) to help frame our work within the broader context of non-perennial streams. We also included additional language highlighting flashy and ephemeral streams (revised lines 16-17; 18; 55). We continue to frame our paper in terms of the existing literature on arid stream hydrology by discussing gaging and modeling challenges and citing 2 new references (Costigan et al., 2017; Merritt et al., 2021) (Lines 67-70). Just before our objectives, we emphasize management challenges of ephemeral streams as a motivator behind our paper (Lines 113-114). In the Methods section, we make a connection between our no flow metric and the literature (Lines 67-68). In our Introduction and Discussion, we add information highlighting the uncertainty of hydrologic methods used in HMC, and we compare them to methods used by other sophisticated stream classification approaches (Lines 211-213; 429-430; 431-435). In doing so, we add 3 new references (Knoben et al., 2019; Ehret et al., 2020; de Vos et al., 2010) and place our method in better context of

model error and existing classification methods that handle data limitations.

***In the revision, please format your author's response as a point-by-point response to the original reviews. I.e. each sentence or point from the reviewer should be copied over word-for-word, with a standalone description of how it was addressed. The current format makes it difficult to ascertain whether all points were addressed.***

We have reformatted our response as the Editor requested so that it is easier to follow.

**Referee 1 general comments:**

***Thank you for the opportunity to review this paper, it was an engaging exercise. Overall this is a well-written paper that presents a seemingly novel approach to stream classification based on hydrologic model error. My remaining concern lies in the lack of literature/background on the physical hydrology and stream classification of flashy and ephemeral streams. While the authors covered the recent classifications of CA including their region of interest, given the focus of this journal and their study aims (specifically the connection they try to make with the PUB initiative), I encourage the authors to include some discussion of the distinct hydrologic processes and rapidly increasing research focus on ephemeral streams more broadly.***

Copied from Editor's first general comment:

We have endeavored to substantially engage with the reviewer comments and provide thorough and robust responses, especially in regard to the general comments provided by Reviewer 1. We have added a new paragraph at the beginning of the Introduction focused on physical hydrologic processes of non-perennial streams (Lines 37-48). We cited 6 new references (Skoulikidis et al., 2017; Stubbington et al., 2018; Datry et al., 2014; Tooth, 2000; Gannon et al., 2022; Hammond et. al, 2020) to help frame our work within the broader context of non-perennial streams. We also included additional language highlighting flashy and ephemeral streams (revised lines 16-17; 18; 55). We continue to frame our paper in terms of the existing literature on arid stream hydrology by discussing gaging and modeling challenges and citing 2 new references (Costigan et al., 2017; Merritt et al., 2021) (Lines 67-70). Just before our objectives, we emphasize management challenges of ephemeral streams as a motivator behind our paper (Lines 113-114). In the Methods section, we make a connection between our no flow metric and the literature (Lines 67-68).

***Further, while modeling ungaged basins and facilitating development of e-flow criteria are related, the aim of the models and specific flow characteristics / processes handled may differ and this should be clearly articulated. At this point the selection of the 2 flow metrics used in the inductive classification feels insufficiently justified as 'best current practices', given the numerous metrics that have been successfully used to describe various aspects of ephemeral/flashy stream flow regimes (See Merrit et al 2021 and references therein as a starting point). This then sets up a somewhat uneven comparison, although I am convinced by the end that the HMC approach is complementary.***

We agree that the selection of the 2 flow metrics was presented in a way that feels insufficient as "best current practices", so we have revised our wording on why the metrics were chosen. Ultimately, this

paper is part of a suite of papers on developing environmental flow criteria in So. CA using the ELOHA approach. Because this paper builds on previous research in the study region, we think it is important to emphasize the study region and preceding studies. As such, we have rewritten Section 2.2.1 in the Methods to bolster our rationale behind using the 2 flow metrics (Lines 154-160). We reword "best current practices" to "existing approaches" in our objectives (Line 118). We more clearly articulate differences between modeling general hydrology and modeling environmental flows (Lines 94-96).

*I would also like to see some background and discussion on past studies tacking hydrologic model error and parameter clustering/regression techniques as a way to handle data limitations and infer watershed similarities and differences to place this method in context (e.g. Ehret et al 2020; De Vos et al 2010; Knoben et al 2020; Beven et al 2020).*

Copied from Editor's first general comment:

In our Introduction and Discussion, we add information highlighting the uncertainty of hydrologic methods used in HMC, and we compare them to methods used by other sophisticated stream classification approaches (Lines 211-213; 429-430; 431-435). In doing so, we add 3 new references (Knoben et al., 2019; Ehret et al., 2020; de Vos et al., 2010) and place our method in better context of model error and existing classification methods that handle data limitations.

*In summary, this paper would be more compelling if framed in terms of the existing literature on arid stream hydrology and hydrologic modeling/ flow regionalization and the study region were introduced later on in the context of an application, rather than as a singular case study.*

Per our responses to Reviewer 1's general comments, we have reframed our paper to be more in line with existing literature on non-perennial stream hydrology and modeling uncertainty, while still emphasizing the study region and all its supporting research.

**Referee 1 specific comments:**

*L55: Not quite accurate. Suggest to change to: Lane et al. (2017) grouped unimpaired gages based on their natural streamflow regime before using watershed characteristics to predict the flow type of ungaged reaches.*

Thank you for pointing out our mistake. We have corrected our interpretation of Lane et al., 2017 (Lines 73-74).

*L60 – Define acronyms (CA, So. CA) before using; Change separate to separation.*

Good point. We defined all acronyms prior to using them (Lines 48, 55, 71, etc.). We changed "separate" to "separation" (Line 78).

*Fig 1 – the county delineations in this map seem unnecessary, and it may help to instead add a few major cities as landmarks for the gage locations. Furthermore, to be accessible to a non-CA or -US*

*audience, having the inset map above just show a floating CA rather than the western US seems confusing.*

We substantially changed Figure 1 to eliminate county delineations, added the cities of Santa Barbara, Los Angeles, and San Diego, and changed the inset to show the western US (Line 147).

*L153 – Suggest to specify that these studies used 'daily average streamflow'*

We specified "daily average streamflow" (Line 176).

*L155 – This paragraph could benefit from some description of what each of these methods/codes actually do, and why this approach was selected. Simply stating a list of indices and packages feels insufficient, as multivariate data analysis is a complex and nuanced process. I also recommend to remove mention of the specific R packages used and simply share a link to the code repository at the end of the paper to support repeatability, a critical part of hydrologic modeling, and make the paper easier to read.*

We agree that a list of indices and packages could be further developed, so we rewrote the final paragraph in Section 2.2.1 to provide more detail about methods and scripts for stream classification (Lines 180-186). We also removed mention of specific R packages from the main text.

*L160 – Similarly to the above comment, 'removing highly correlated metrics' is a subjective undertaking and some discussion of how this was done would increase transparency/repeatability.*

The following paragraph in Section 2.2.2 was rewritten to provide further discussion of how highly correlated metrics were removed and to remove mention of specific R packages (revised lines 175-178). We removed mention of specific R packages from the following paragraph in Section 2.2.3 (revised line 183).

*Figure 2 – I challenge the authors to try to develop a more intuitive flowchart to visually depict this somewhat complex process, including a visual of the example provided in the text below. See Figure 1 in Ehret et al 2020 as one example.*

We agree that Figure 2 was not an intuitive depiction of HMC. Thank you to Reviewer 1 for pointing us in the direction of Ehret et al 2020. We completely redeveloped Figure 2 into a more intuitive flowchart to provide a better visual tool for understanding this complex process (Line 221)

*L320 – Do you provide the model performance values to justify 'successful calibration'*

Thank you for pointing out our use of the phrase "successful calibration". We believe that this is subjective and inappropriate wording, so we reworded the reporting of our results to clarify model calibration was accurate, not "successful" (revised lines 350-351). We deemed calibration as "accurate" due to average percent errors under 1%.

**Referee 2 general comments:**

*thank you very much for this work. I think the work is interesting but i have some suggestions that i hope help to improve the manuscript. My main concern is about the recent literature review in ungauged catchments. I miss some recent references. For example, the authors might want to have a look at the work from Almeida et al., Le Vine et al., Addor et al., Kratzert et al.., Fenicia et al., Kavetski et al..*

We agree with Reviewer 2 that we needed to bolster our background on ungaged catchments. As such, we augmented our background literature related to ungaged catchments (Lines 104-106; 107-109). We discuss regression, random forest, and neural network models while adding 5 new references (Abdulla and Lettenmaier, 1997; Seibert, 1999; Yokoo et al., 2001; Prieto et al., 2019; Kratzert et al., 2018).

**Referee 2 specific comments:**

*page 3: "singular streamflow metrics". Please, clarify/re-word*

"Singular streamflow metrics" was unintentionally plural, so we made it single (Line 89).

*page 3: what about transferring hydrological index?*

Good point. We added transferring by hydrologic similarity (Line 100).

*page 3: "network of models". Please, clarify/reword*

We agree that "network of models" is confusing and so we reworded to "many models" (Line 98) and changed "network" to "study area" (Line 99).

*page 3: "process-based". Just a suggestion, but you might want to specify conceptual instead of process based.*

Good point. We specified that we used "conceptual" models and not just process-based (Lines 89-90; 429-430).

*page 6: equations (1) and (2) and the remaining equations in the manuscript. Please, use proper mathematical notation (eg matrices in bold, etc)*

We ensured all equations use proper mathematical notation required by HESS (Lines 164, 169, etc.)

*page 6: "1cfs". Could you please explain a bit why this specific number is chosen*

Certainly. This will likely confuse other readers. We provided more detail on why we chose 1 cfs as the threshold for a dry stream (Lines 170-171).

*page 6: "weighted classical (metric) multidimensional scaling" could you please provide some brief description, same for the c-index, Dunn, McClain and Silhouette.*

Copied from Reviewer 1's fifth specific comment:

We agree that a list of indices and packages could be further developed, so we rewrote the final paragraph in Section 2.2.1 to provide more detail about methods and scripts for stream classification (Lines 180-186). We also removed mention of specific R packages from the main text.

*page 7: "calibrated paramters inherentely have greater uncertainty than directly calculated parameters". Please, explain a bit explicitly what you are trying to say.*

Sure. We clarified the statement by explaining how calibrated parameters are often difficult to define physically and frequently lack data needed for their direct calculation.

*page 7: "reciprocating model accuracy". Please, explain a bit explicitly what you are trying to say.*

Sure. We explained that reciprocating model accuracy means the accuracy of a model when calibrated parameters from a different model are donated to it, and vice versa (Lines 192-194).

*page 9: I think a flow chart with the specific steps would be helpful*

Copied from Reviewer 1's seventh specific comment:

We agree that Figure 2 was not a good visual depiction of HMC. Thank you to Reviewer 1 for pointing us in the direction of Ehret et al 2020. We completely redeveloped Figure 2 into a more intuitive flowchart to provide a better visual tool for understanding this complex process (Line 221)

***page 13: "succesful". I suggest you to elaborate this more, what do you mean by succesful? why 1% whas choosen and how?***

Copied from Reviewer 1's eighth and final specific comment:

Thank you for pointing out our use of the phrase "successful calibration". We believe that this is subjective and inappropriate wording, so we reworded the reporting of our results to clarify model calibration was accurate, not "successful" (revised lines 350-351). We deemed calibration as "accurate" due to average percent errors under 1%,

1% was not chosen for a specific reason but is a very small error indicative of accurate calibration.

***references: please, check for the DOIs***

Thank you for this suggestion. We added DOIs to citations.